# Photostability of Oil-Coated and Stain-Coated Acetylated Hornbeam Wood against Natural Weather and Artificial Aging

Fanni Fodor *, Miklós Bak and Róbert Németh

Institute of Wood Technology and Technical Sciences, Faculty of Wood Engineering and Creative Industries, University of Sopron, H-9400 Sopron, Hungary; bak.miklos@uni-sopron.hu (M.B.); nemeth.robert@uni-sopron.hu (R.N.)
* Correspondence: fodor.fanni@uni-sopron.hu

**Abstract:** Nine different environmentally friendly coatings were tested on natural and acetylated hornbeam wood, during natural weather exposure and xenon lamp irradiation. The coating performance of acetylated hornbeam, and the photostability properties of tested coatings were evaluated to offer suggestions regarding suitable and less-suitable coatings for the exterior use of acetylated hornbeam. On the one hand, acetylation decreased the coating absorbance of hornbeam. On the other hand, it made the wood more durable and dimensionally stable, all of which influences the outdoor performance of acetylated hornbeam. The color of acetylated hornbeam is not photostable; it brightens during photodegradation, and greys after leaching. For long-lasting color, acetylated hornbeam should be coated with dark pigmented stains and maintained regularly. Fungal degradation and cracking did not occur, but the wood is just as susceptible to wasp stripping as untreated hornbeam. In this study, 200-h-long xenon lamp irradiation resulted in a color similar to that caused by 1 month of weather exposure (April to May 2018, Sopron, Hungary).

**Keywords:** acetylation; coating; color; photostability; weather; wood modification; xenon

## 1. Introduction

Color inhomogeneity is an important wood characteristic. Usually, the color hue varies between dark brown and bright yellow, but it can also be enhanced for a more intensive color appearance [1]. The overall appearance also depends on the grain figure, the contrast between earlywood and latewood, annual growth ring width, pith rays, and ray flecks. Despite being rather inhomogeneous among species and within wood itself, the contrast and deviation between wooden elements provide little stimuli and no fatigue. Hence, human perception tends to consider wood as "warm", "natural", "beautiful", "soft", and "elegant" [2].

Hornbeam has greyish-white sapwood and heartwood, which turn yellowish or occasionally dark brown when exposed to air due to oxidation. Annual rings are wavy on the cross-section. The wood has small, diffuse pores and a structure with relatively small porosity. There are two types of rays: the narrow, 1–2 cell-wide rays are invisible; the 3–4 cell-wide aggregate rays, mixed with libriform fibers, are well discernible with the unaided eye as darker flecks on the radial and as small stripes on the tangential section. Pith flecks are also sometimes present [3].

The color range of various domestic European hardwood species comprises about 0.6% of the whole CIE L*a*b* color space [1]. Much research has been performed to determine categories of color differences to grade wood. If ΔE* reaches or exceeds 2, the color difference is perceivable [4–9]. Bio-based materials are sustainable and have a wide range of appearances, unique properties, and natural beauty, which is favorable for users, thus it is important to guarantee the due aesthetical performance. However, specific local and cultural circumstances also need to be considered [10].

This research has been carried out in Sopron, Hungary. Hungary's climate is very erratic because it is situated in between three climatic zones: oceanic climate with less varying temperature and more evenly dispersed precipitation; continental climate with more extreme temperature and relatively moderate precipitation; and a Mediterranean effect with dry weather in summer, and wet weather in winter. These factors make Hungary's weather variable despite its lower altitudes and relatively small extent. Hungary has a continental climate, with hot summers with low overall humidity levels but frequent showers and frigid to cold snowy winters. Sopron is in the moderately cool–moderately dry climatic region [11]. These characteristics suggest that wood is likely to grey, mold, and crack in a short amount of time without proper treatment.

Various physical, chemical, and biological processes occur when wood is exposed to the weather. A variety of factors cause these processes, including moisture (precipitation, moisture in soil), UV radiation (sunlight), frequent temperature and relative humidity changes, thermal impact, wind, sand, and degradation by fungi and insects [12–14]. Degradation caused by weathering usually affects depths of 0.5–2.5 mm [15].

During exposure, the lightness (L*) can have some initial increment, but after some time its value decreases. In addition, a* and b* can have some initial increment during yellowing, but both eventually decrease as the surface starts to grey [16,17].

Wood can also be exposed to artificial ageing. Laboratories can reproduce photodegradation using irradiating lamps (xenon, mercury vapor, fluorescence) to accelerate the degradation rate using shorter wavelengths. The advantages of this form of photodegradation include measurement reproducibility, constant settings, and shorter testing times. Testing time is shortened by wavelengths that are shorter than the UV spectrum hitting the earth's surface (290–400 nm) [17,18].

If the rate of deterioration is plotted as a function of time, different patterns can be defined: the concave pattern, the convex pattern, the "S"-shaped pattern, and the linear pattern [19].

The grey color of photodegraded wood is also associated with fungi. Some fungi grow on weathered wood, for example to metabolize photodegradation products. These fungi are in the division Ascomycota, and are usually one or more of the following four species: *Aureobasidium pullulans*, *Hormonema dematioides*, *Epicoccum nigrum,* and *Phoma* spp. Other species belonging to the *Clasosporium* and *Scerophoma* can also be found. They are called "black stain fungi" and they give a grey coloration on delignified wood surfaces, and black or brown coloration on lignified wood [20]. Wood modification and coating can be used to protect wood from performance loss caused by photodegradation and weathering [21]. Coating is widespread because it is more convenient and less expensive. On the other hand, its disadvantages include leachability, susceptibility to mechanical damage, and the need for renewing its application. Coating usually performs better on low-density wood than on high-density wood because of the lower swelling–shrinking rate. A higher swelling–shrinking rate and higher stresses can occur between earlywood and latewood, which can create stresses in the coating of high-density wood and lead to coating failure [22].

The ideal finish allows the natural visual characteristics to be visible, minimizes UV discoloration, reduces water exchange between the atmosphere and the material, and accommodates the dimensional changes of the material to reduce cracking and peeling from the wood surface for the longest possible time, thereby reducing maintenance expense.

Coatings can be film-forming or non-film forming; this study tested the latter. Non-film forming coatings are stains (pigmented or clear), which resist UV and water penetration while highlighting some visual characteristics but are less effective than film-forming paints. Easy but more frequent maintenance is required [21]. Penetrating oil finishes can reduce dimensional changes, thereby decreasing cracking and checking; however, the oil degrades over time, and the protective characteristics of the coating are diminished. After the coating has either peeled or been destroyed, the natural process of color change will begin to occur [20,22].

Oil-based coatings are mainly triglycerides including different fatty acids in esters. Wood impregnating oils and binders in paint are usually drying oil or semi-drying oils. Vegetable oils can be efficiently used in wood protection for their antifungal, antibacterial, water-repellency, and UV stability. The downside of oil coatings is that they only fill the wood cavities by capillarity, without chemically bonding to the cell walls. Thus, it is necessary to ensure high oil retention in order to acquire the required protection. Drying oils (often mixed with oil-modified polyester resins or alkyds) first undergo an auto-oxidation reaction under air exposure, form a structural tri-dimensional network through cross-linking, then a radical chain reaction takes place. Some of the highly conjugated polymeric networks are UV-sensitive–they are prone to yellowing and discoloration. Higher amounts of linoleic acid with two double bonds increase the speed of drying, but also the rate of photochemical degradation. Linseed oil contains 48–60 w% linolenic acid, 14–19 w% linoleic acid, 14–24 w% oleic acid, 6–7 w% palmitic acid, and 3–6 w% stearic acid. Ether bridges increase the resistance to water and alkali. Linseed, tall, orange, soybean, nut, and hemp oil are efficient for wood preservation, mainly against decay and termites; while tall, linseed, and tung oil are very efficient against water uptake [23].

Natural and modified natural resins are also important components of natural wood coatings. Plant resins are usually synthesized by coniferous plants and contain terpenes and derivatives. They are transparent and have an amorphous chemical structure. The most frequently used resins in coatings are shellac, colophony, and dammar [24]. The rate of surface degradation is typically related to UV intensity, time of exposure, wood species, and climatic factors [25–28]. In the early stages of weathering, dark wood tends to become light, and light wood tends to darken or turn silver [29,30]. In the case of colored coatings, pigments and dyes are also degraded. Some pigments can be more photo-resistant than the binder, so they become non-bound to each other after the destruction of the binder, which leads to "chalking" [20]. Due to chemical changes during photodegradation, the coating can become more brittle, cracks can appear, and the coating film becomes thinner. Wood and coatings age according to the same phenomena [20,21]. With clear coatings, the loss of protection is due to delamination rather than the destruction of the polymeric matrices [31]. Exposing the coating to water flow causes the leaching of pigments, hydrolysis of organic compounds, fungal degradation, and wood dimensional changes, which reduce coating performance and service life [20].

Acetylation is a chemical modification method used to bond a reactive, less hydrophilic, simple chemical (acetyl group) to a reactive part (hydroxyl group) of a wood polymeric constituent (lignin, hemicelluloses, cellulose). The rate of acetylation–the indication of successful modification–is usually expressed as the weight percentage gain (WPG). The lower shrinking and swelling rate enables a longer-lasting coating film. The higher the WPG, the more durable the coating of acetylated wood will be. Wood-decaying fungi can colonize acetylated wood, as hyphae can be present in its microstructure, but weight loss does not occur or only to a low extent. This means that acetylated wood is not toxic to the fungus, but the nutrients are not sufficient for fungal growth at higher WPG levels [32].

It is reported that acetylation increases the weather (light and moisture) resistance of wood [32–40], noting that the photostability increases at higher WPG levels [35]. However, acetylated wood only shows initial stability against UV radiation; later, it begins to fade and grey [37–39,41–46]. This is also true for Accoya® Radiata pine [47].

The color change caused by photodegradation is influenced by the structural change of extractives too. The degradation of lignin is hindered due to acetylation and the increased moisture resistance, and dimensional stability restrains the photodegradation mechanism of wood [48]. On the other hand, acetylating the phenolic OH groups, which retard the formation of quinones, reduces this protective mechanism [13,49].

Water-based paints penetrate less and a form thinner film on acetylated wood than on untreated wood. Flaking can be a problem because of increased acidity and buffering capacity [50]. Case studies have shown that while outdoor paints and stains need to be repainted every 2–6 years, Accoya requires 10–12 years [51]. Various case studies addressed

the topic of coating acetylated wood and have highlighted the durability of alkyd and acryl coating systems. The maintenance time or service life of coated acetylated wood can be three or four times longer with appropriate surface preparation, surface treatment, and a well-chosen coating system [52–55]. Alkyd and acrylic stains were tested at 13 years of exposure, with the best results being the alkyd primary and acrylic topcoats, which did not need to be repainted after 13 years. If a suitable coating system is used, it may be sufficient to maintain the coating every 10 years or even less frequently [54].

The present work tested various environmentally friendly, waterborne, oil-based coatings on natural and acetylated hornbeam during natural weather exposure and xenon lamp irradiation. Then, the coating performance of acetylated hornbeam and the photostability properties of tested coatings were evaluated to determine suitable and less-suitable coatings for exterior use of acetylated hornbeam.

## 2. Materials and Methods

Samples of untreated and industrially acetylated hornbeam (*Carpinus betulus* L.) (average WPG 15.3%) were manufactured with dimensions of $20 \times 20 \times 100$ mm$^3$ (t $\times$ w $\times$ l). The average density of hornbeam and acetylated hornbeam were $745 \pm 42$ kg/m$^3$ and $795 \pm 55$ kg/m$^3$, respectively. Acetylation was carried out the same way as in previous work [56].

There were nine different oils and stains tested in this research: Uncoated (marked as 0 throughout), Oli-Natura oil-colorless (OO0), Auro stain-colorless (AS0), -umbra (ASU), Biopin stain-colorless (BS0).-Swedish red (BSR),-palisander (BSP), Auro oil-teak (AOT), Biopin oil-teak (BOT), and -red (BOR) (Tables 1 and 2). The finishes were applied using a paintbrush according to the manufacturer's instructions. The products originated from three different companies: Stoki Ltd. (Oli-Natura), Sixay Furniture Ltd. (Auro), and Orange Ltd. (Biopin). Overall, 10 samples were made of each type of coating and wood. Of these, five samples were used for weather exposure and five for irradiation. The sample weight was determined before and after coating to define weight gain (%, g/m$^2$) and observe the uptake difference between non-acetylated and acetylated wood.

**Table 1.** Composition of tested coatings.

| Product Name, Type and Marking | Ingredients |
| --- | --- |
| Oli-Natura Yacht and Teak Oil<br>–　colorless (OO0) | Modified vegetable oils such as linseed oil; isoparaffinic hydrocarbon of pharmaceutical quality; lead-free drying agents, non-fading earth and mineral pigments, micronized titan oxide. |
| Auro Garden Furniture Oil No. 102<br>–　teak (AOT) | Fatty acids, lecithin, linseed oil, mineral pigments, orange terpene, silica, tung oil, drying agents (cobalt-free). |
| Auro Wood stain No. 160<br>–　colorless (AS0)<br>–　umbra (ASU) | Water, linseed oil, colophonium glycerine ester with organic acids (as ammonium soaps), mineral fillers and pigments, surfactants made from rapeseed and Ricinus oil, silicic acid, dryers (cobalt-free), castor stand oil, sunflower oil, cellulose, fatty acids. |
| Biopin Weather Protection Stain<br>–　colorless (BS0)<br>–　palisander (BSP)<br>–　Swedish red (BSR) | Binding agents (polymer made from natural oils, fatty acids, and resins), solvent (water), pigments and fillers (earth and mineral pigments), additives (vegetable-based emulsifier, lead and barium-free stabilizers and drying agents, methyl cellulose, film treaters (3-iodine-2-propynyl butylcarbamate (max. 0.2%), and 2-octyl-2H-isothiazol-3-one (<0.01%)). |
| Biopin Terrace Oil<br>–　teak (BOT) | Binding agents (polymer made from natural oils, fatty acids, and resins), solvent (isoparaffin), pigments (mineral pigments), additives (silicic acid, lead-free drying agent). |
| Biopin Natural Impregnating Oil and 20% K.O.S. Natural Pigment Paste<br>–　red (BOR) | Binding agents (polymer made from natural oils, fatty acids, and resins), solvent (isoaliphate), additives (lead and barium-free drying agents), and non-toxic natural mineral pigments prepared in vegetable oil. |

**Table 2.** Markings and important properties of tested coatings.

| Marking | Volume Solids (%) | Density (g/cm³) | Drying Time (Hours) | Full Hardness (Day) | Number of Layers | Application Amount (g/m²) |
|---------|-------------------|-----------------|---------------------|---------------------|------------------|--------------------------|
| OO0 | 68.39 | 0.875 | 1–2 | 2–3 | 2 | 40–80 |
| AOT | 90.53 | 0.95 | 24 | 28 | 2 | 38 |
| AS0 | 12.78 | 1.065 | 24 | 28 | 3 | 75–96 |
| ASU | 31.96 | 1.065 | 24 | 21 | 3 | 75–96 |
| BS0 | 58.59 | 0.98 | 24 | 14 | 3 | 69–83 |
| BSP | 50.97 | 0.98 | 24 | 14 | 3 | 69–83 |
| BSR | 59.15 | 0.98 | 24 | 14 | 3 | 69–83 |
| BOT | 72.94 | 0.84 | 4–6 | 14 | 3 | 55–105 |
| BOR | 50.26 | 0.86 | 6–8 | 14 | 3 | 60–69 |

The UV irradiation test was performed in an artificial ageing chamber. The test used two xenon lamps with 482 W/m² average radiant power density. The lamps were 64 cm above the samples and the equipment temperature was set to a maximum of 50 °C.

The samples were exposed to weathering without soil contact on the roof of The Central Library of the University of Sopron (47°40'54.3" N 16°34'40.6" E) in April 2018 (Figure 1). They were placed on a wood stand tilted at 45 degrees. The time of exposure was 24 months. The weather parameters of the testing field were obtained from the Department of Ecology and Bioclimatology of the University of Sopron. These include average and maximum monthly temperature, monthly precipitation, number of days with precipitation above 0.25 mm, sunshine duration per month, solar irradiance per month, and monthly relative humidity. The Scheffer Climate Index (SCI) was also calculated, which was proposed to estimate the decay hazard by geographic location within the conterminous United States for wood exposed above ground to exterior conditions [57]. This is calculated from local weather data, using the mean monthly temperature and mean number of days with at least 0.25 mm of precipitation over the exposure period. An index less than 35 represents the least favorable conditions for decay; 30 to 65, intermediately favorable conditions; and greater than 65, conditions most conducive to decay. As a metric by which relative hazard can be compared between geographic locations, the Scheffer index is not intended to predict decay propagation rate nor time to failure in specific constructions [58]. This particularity is not necessarily a limitation of the approach, but it has been debated. However, the SCI is still the most frequently used index of its kind for estimating the relative climate-induced decay hazard of geographical locations [59].

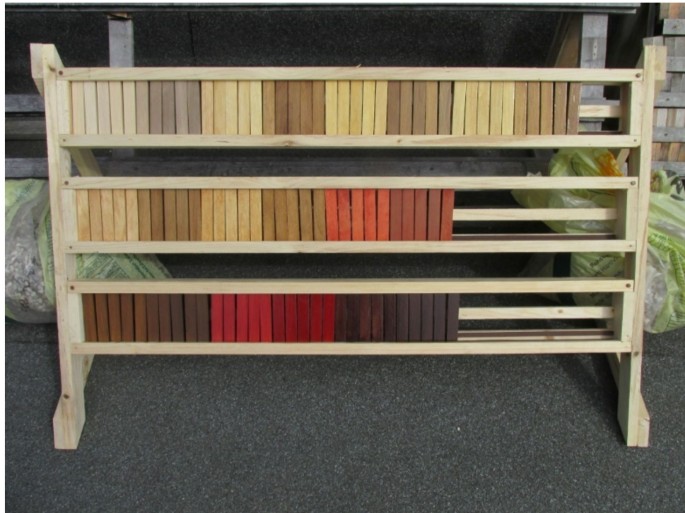

**Figure 1.** Weather exposure of untreated and acetylated hornbeam wood coated with various oils and stains.

Each sample had four measuring points, which equals 20 for each type per test. A Konica Minolta 2600d spectrophotometer determined the color. The colorimeter's sensor head was 8 mm. The color was measured and calculated based on the D65 illuminant and 10° standard observer. The color coordinates were calculated according to ISO 11664:2019.

The color measurement and visual inspection of the xenon lamp-irradiated samples were completed after 0–5–10–20–30–60–120–200 h of irradiation. The color measurement and visual inspection of the weathered samples were conducted at the beginning, then every 10 days (less frequently later) in their current state without conditioning. A color chart was also created by converting CIE L*a*b* coordinates to values of RGB (Red Green Blue) for a video display screen [60]. The conversion was based on the sRGB (Standard Red Green Blue) definitions of the light emitted from the display screen. Photos and scans were taken of the samples in their current state to compare visual appearance.

## 3. Results and Discussion

### 3.1. Coating

According to the weight gain results, hornbeam absorbed less coating after acetylation (Figure 2). This means that the coating penetrated the wood less and formed thinner film compared to natural hornbeam. In the cases of AS0, ASU, BOT, and BOR, the weight gain did not reach the application amount given by the manufacturer, which can be explained by its dense and less porous structure.

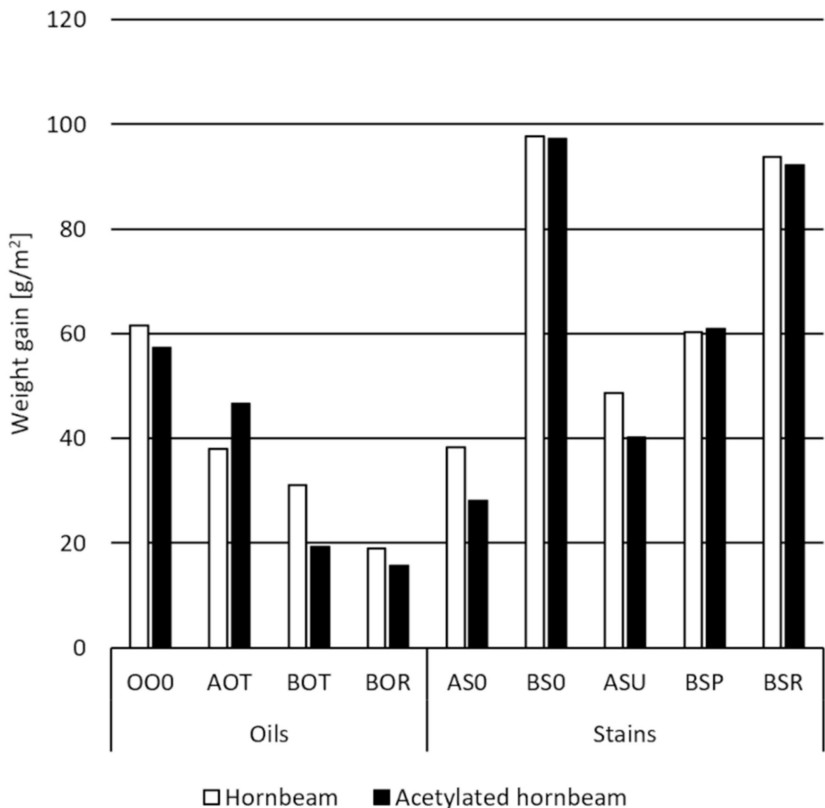

**Figure 2.** Weight gain of natural and acetylated hornbeam after coating, at 20 °C temperature, 65% relative humidity.

Acetylated hornbeam takes up less moisture, which indicates lower uptake of water-borne coatings. After 49 days, the maximum water uptake of hornbeam was 5513 $g/m^2$, which decreased to 4559 $g/m^2$ after acetylation [56].

In wettings studies, the contact angle of hornbeam increased from 43–44° to 61–62°, and the surface energy decreased from 55 to 64 $mJ/m^2$, which makes acetylated hornbeam less wettable and less coatable [61].

The coating performance of hornbeam also depends on its chemical properties. After acetylation, its pH decreased from 5.11 to 4.73, and its buffering capacity increased from 1.11 to 2.15 mg/g [62], which can lead to coating flaking [50].

The natural color became darker with colorless oils and stains. The wood also obtained a more characteristic grain figure. L* decreased; a* and b* increased. The grain structure was less prominent after coating with pigmented oils and stains, especially with thicker films like ASU, BSR, and BSP (Table 3).

**Table 3.** Color coordinates of hornbeam and acetylated hornbeam, uncoated and coated with various oils and stains.

| Marking | Hornbeam | | | Acetylated Hornbeam | | |
|---------|------|------|------|------|------|------|
| | L * | a * | b * | L * | a * | b * |
| 0 | 78.73 | 4.06 | 19.70 | 58.31 | 5.38 | 15.94 |
| OO0 | 73.09 | 7.39 | 29.48 | 47.36 | 10.48 | 23.90 |
| AS0 | 76.06 | 5.55 | 32.66 | 49.40 | 11.42 | 25.00 |
| BS0 | 75.56 | 5.41 | 29.96 | 49.91 | 9.63 | 24.96 |
| AOT | 67.34 | 11.69 | 33.42 | 50.46 | 8.52 | 22.87 |
| ASU | 40.57 | 10.62 | 17.53 | 36.87 | 8.09 | 11.76 |
| BSP | 32.87 | 8.17 | 6.18 | 30.31 | 6.31 | 5.06 |
| BSR | 38.50 | 28.22 | 17.81 | 35.50 | 23.74 | 15.10 |
| BOT | 69.59 | 9.94 | 27.14 | 47.35 | 9.87 | 21.18 |
| BOR | 49.60 | 31.08 | 27.86 | 42.19 | 21.19 | 21.07 |

### 3.2. Xenon Lamp Irradiation

After 200 h of xenon lamp irradiation, OO0 experienced the most prominent ΔL*. Conversely, BS0 had the smallest ΔL* among the colorless coatings, while dark, pigmented stains had the smallest ΔL* of all coatings tested (ASU, BSP, BSR) (Figure 3). Similar results can be found in literature [29,30], where dark woods brighten and light woods darken during UV exposure.

OO0, AOT, and BOT had the biggest Δa*. BSP had a great increment in a* in acetylated wood. BS0 had the smallest Δa* among the colorless coatings, while dark, pigmented stains had the smallest of all coatings tested (ASU, BSR, BOR). Coating with teak oil showed results that were similar to the control (Figure 4).

Compared to the other color coordinates, b* changed less consistently. Coating with colorless oils and stains increased Δb* during irradiation. Coating with teak oil showed similar results to the control. Acetylated hornbeam had the lowest Δb*, but a small change was also found in dark, pigmented stains like ASU and BSR (Figure 5).

In every case, the color difference after 200 h of irradiation was perceivable (ΔE* > 2) [9]. ASU, BSR, and BSP achieved the lowest ΔE* in hornbeam and acetylated hornbeam. Hornbeam had a lower ΔE* after irradiation than acetylated hornbeam. The use of non-pigmented oils and stains exhibited the greatest ΔE* after irradiation. Coating with teak-colored oils showed about the same photostability as uncoated wood (Figure 6); it had a concave deterioration pattern (according to [10]).

During irradiation, photo oxidation and cross-linking reactions take place. First, the polymer absorbs photons, which generates free radicals. Then, reactions take place between the oxygen and free radicals, which create peroxy radicals. Peroxy radicals react with the polymer to form a polymer hydroperoxide and a polymer alkyl radical. Following this, hydroperoxides are broken into oxy and hydroxyl radicals. Finally, cross-linking reactions take place between the free radicals [20].

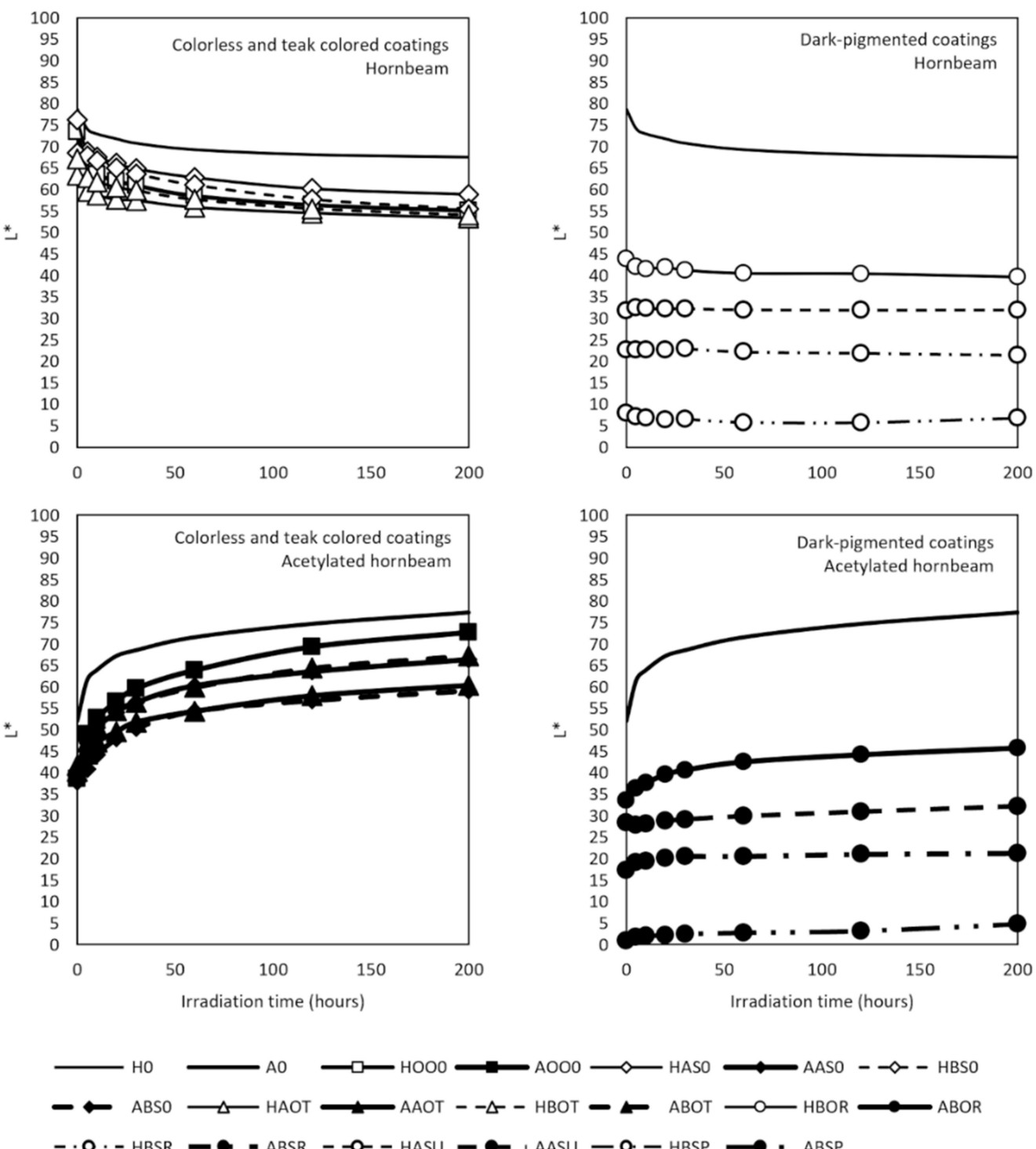

**Figure 3.** Change of lightness (L*) during 200 h of xenon lamp irradiation.

The value of color variation–the color difference calculated compared to the average color–indicated how variable the initial color is. There were no remarkable differences found between the initial color variation of uncoated and coated wood, or untreated and acetylated hornbeam, and all values were lower than 5 (Table 4). After irradiation, the color variation was higher in acetylated hornbeam compared to untreated hornbeam.

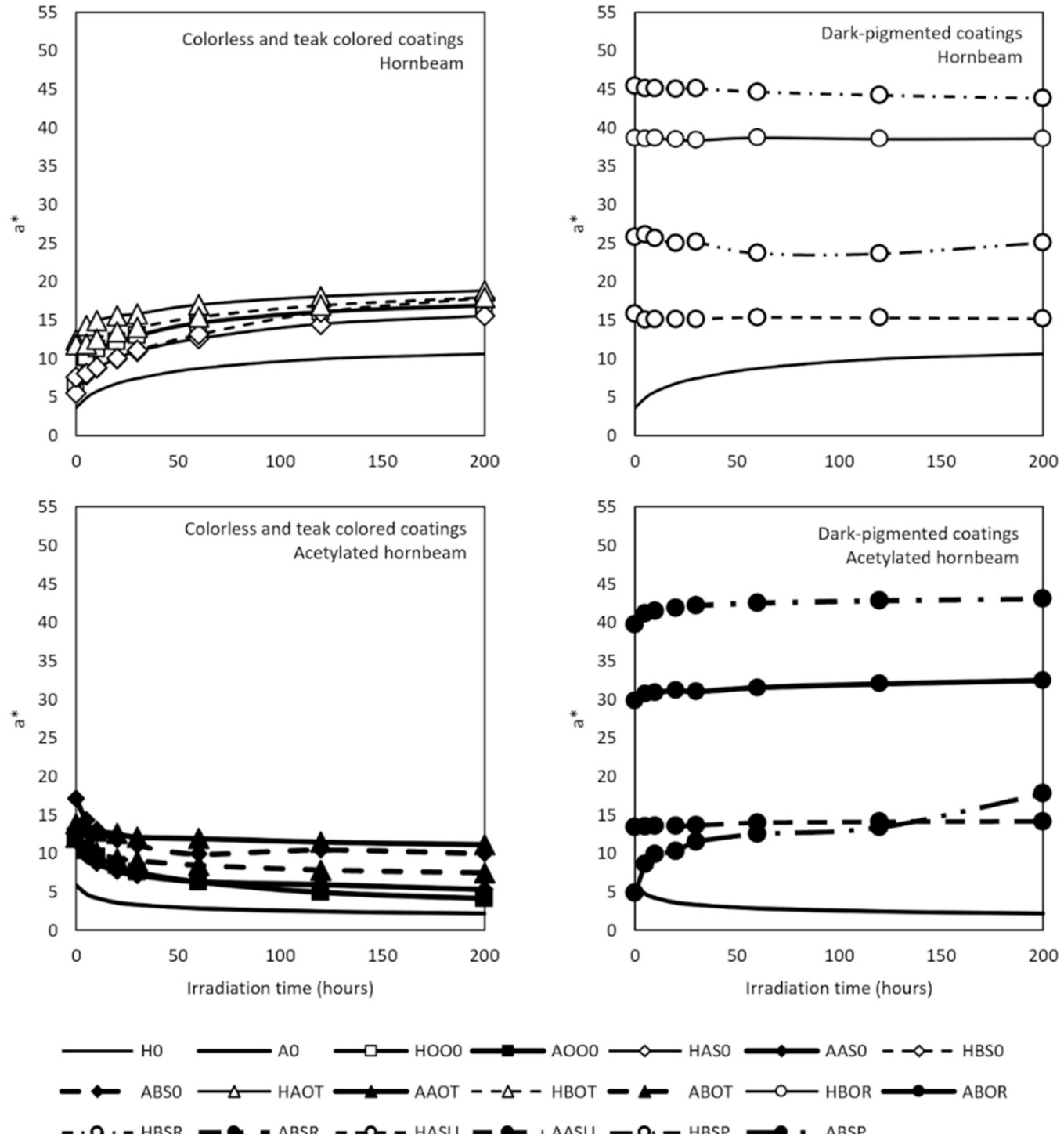

**Figure 4.** Change of red hue (a*) during 200 h of xenon lamp irradiation.

In the case of uncoated, colorless coated (oils and stains), and teak colored samples, hornbeam b* increased as its a* increased, while acetylated hornbeam b* decreased as its a* decreased. On the other hand, dark, pigmented coatings (red, umbra, and palisander) had similar results in hornbeam and acetylated hornbeam: a* and b* decreased or did not change remarkably.

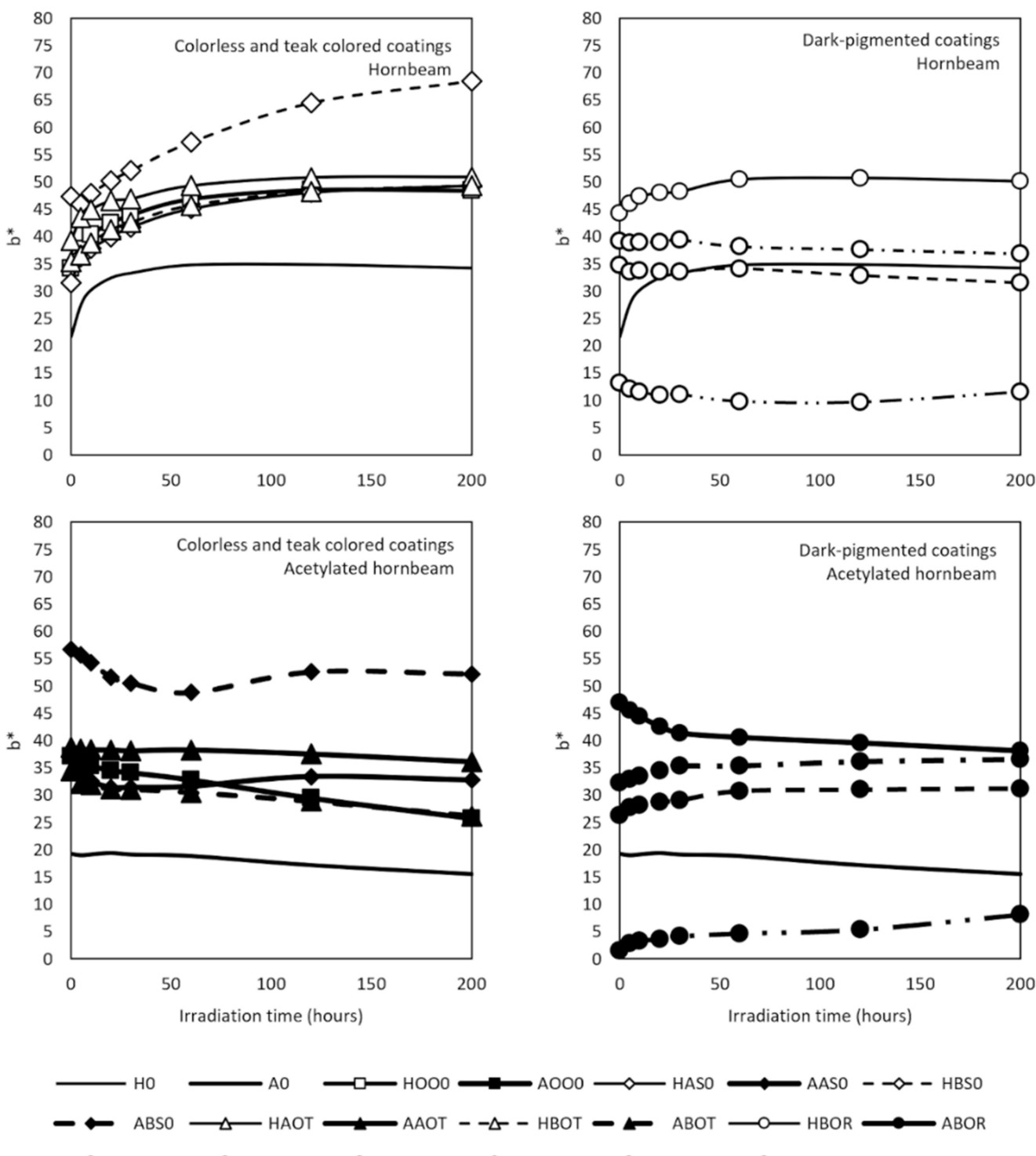

**Figure 5.** Change of yellow hue (b*) during 200 h of xenon lamp irradiation.

Artificial aging only incurs UV light degradation and eliminates the impact of other weather elements and microorganisms. Consequently, the color coordinates mostly increase. Xenon light can simulate the effect of sunlight during weathering, but only at long exposure times, as the yellowing will be about three times more intensive in the short term [17].

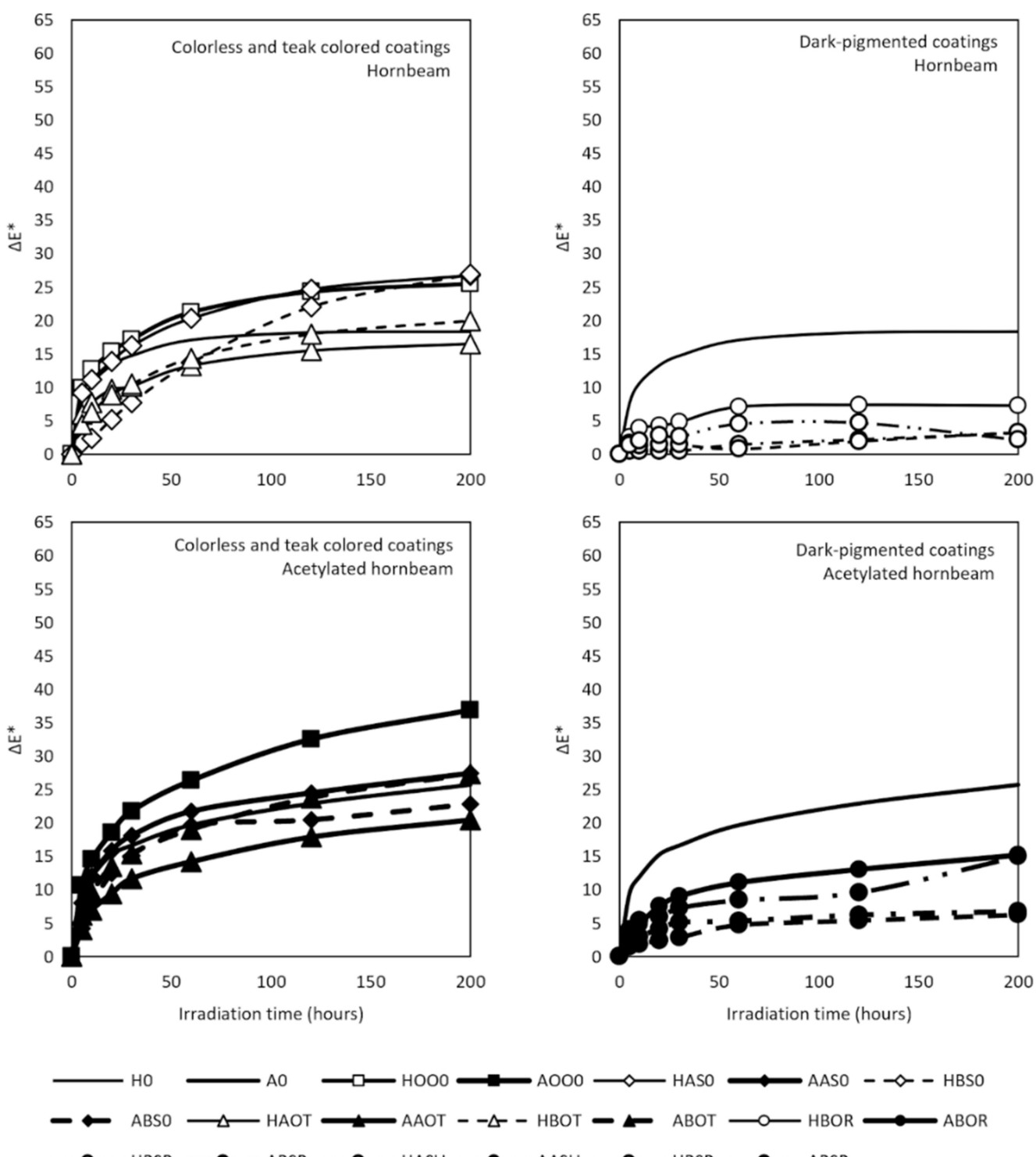

**Figure 6.** Change of color difference (ΔE*) during 200 h of xenon lamp irradiation.

### 3.3. Weather Exposure

Table 5 lists the weather parameters of this site. Compared to hot and dry territories, the Hungarian field probably accelerates decay. This area has a warm and wet summer season (May-September), with mean temperatures between 13–23 °C that range up to 36 °C, and the monthly average precipitation is between 42–154 mm. It has a drier winter season (October–April) with 7–79 mm of average precipitation per month, and mean temperatures between 1–16 °C with a maximum of 29 °C. The average annual rainfall

during the test period was approximately 636 mm, average annual temperature was 13 °C, and the maximum temperature was 36 °C. The relative humidity ranged from 45 to 88%. The direct normal solar annual radiation index at this site is approximately 3.65 kWh/m$^2$/day, representing a moderate UV exposure, with a high likelihood that surface microbial activity (mildew) will occur throughout the experiment.

**Table 4.** Color variation of xenon-lamp-irradiated hornbeam (color difference compared to average color). Average values of 20 measurements are presented.

| Marking | Hornbeam | | Acetylated Hornbeam | |
|---|---|---|---|---|
| | Before | After 200 h | Before | After 200 h |
| 0 | 1.80 | 18.30 | 1.84 | 25.70 |
| OO0 | 4.07 | 25.47 | 3.96 | 36.92 |
| AS0 | 1.49 | 26.78 | 3.12 | 27.41 |
| BS0 | 2.71 | 26.92 | 1.78 | 22.80 |
| AOT | 2.66 | 16.53 | 3.09 | 20.46 |
| BOT | 3.87 | 19.95 | 5.10 | 27.32 |
| BOR | 3.07 | 7.28 | 1.94 | 15.22 |
| BSR | 1.57 | 3.16 | 1.09 | 6.72 |
| ASU | 2.09 | 3.27 | 1.54 | 6.27 |
| BSP | 1.23 | 2.14 | 1.27 | 14.98 |

**Table 5.** Monthly breakdown of average weather parameters.

| 2018–2020 | J | F | M | A | M | J | J | A | S | O | N | D |
|---|---|---|---|---|---|---|---|---|---|---|---|---|
| Temperature (°C) AVG: 13 | 1 | 6 | 8 | 14 | 16 | 22 | 22 | 23 | 17 | 13 | 7 | 3 |
| Precipitation (mm) SUM: 625 AVG: 51 | 20 | 11 | 26 | 23 | 125 | 78 | 61 | 52 | 84 | 32 | 70 | 43 |
| Max temperature AVG: 25 | 13 | 20 | 22 | 27 | 28 | 33 | 35 | 35 | 32 | 24 | 19 | 14 |
| Sunshine duration (hours) SUM: 2548 AVG: 216 | 100 | 173 | 217 | 296 | 243 | 316 | 309 | 297 | 244 | 188 | 89 | 79 |
| Solar irradiance (MJ/m$^2$) SUM: 4895 AVG: 415 | 123 | 241 | 390 | 584 | 585 | 710 | 699 | 618 | 443 | 281 | 123 | 99 |
| Relative humidity (%) AVG: 69 | 81 | 63 | 59 | 55 | 72 | 67 | 61 | 64 | 70 | 77 | 88 | 80 |
| Number of days with precipitation above 0.25 mm SUM: 83 | 1 | 6 | 8 | 14 | 16 | 22 | 22 | 23 | 17 | 13 | 7 | 3 |

AVG: average, J–D: initials of months.

Fungi require oxygen, moderate temperature (25–35 °C), 35–50% of air relative humidity, nutrients, a certain pH, vitamins, and minerals, however water is the main growth factor [20]. This suggests that the summer months were favorable for fungi growth.

According to the ombrothermic diagram (Figure 7), there was a short dry season in February and April (the area below the temperature line and above the precipitation line). Other than that, the wet season–which is the area below the precipitation line and above the temperature line–was typical for the whole year. In the dry seasons, cracking, insect activity, and UV radiation have a greater degrading effect, while in the wet seasons, fungal activity and leaching are more drastic.

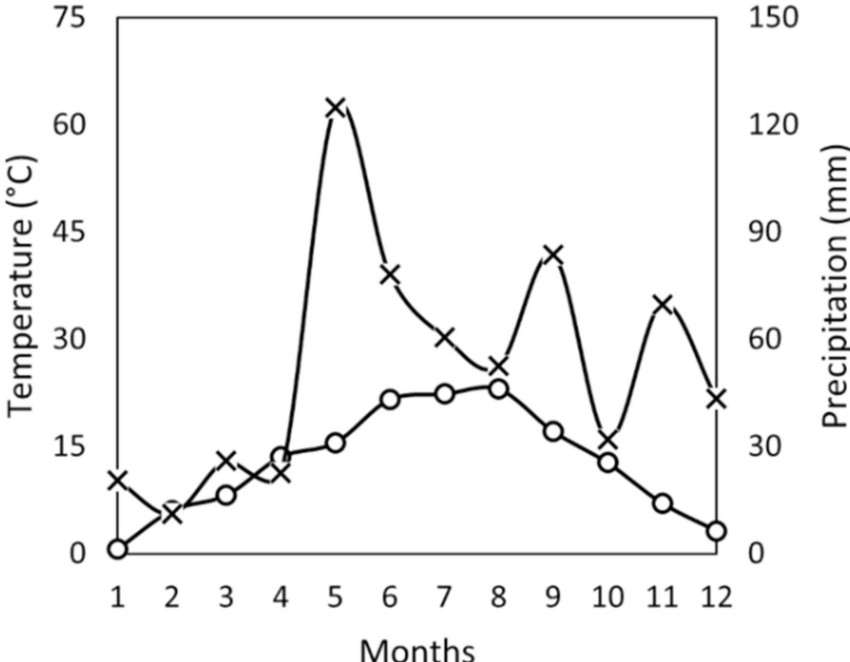

**Figure 7.** Ombrothermic diagram, which summarizes trends in temperature (O) and precipitation (X) of two years of exposure.

The Scheffer Climate Index of this site is 42.1 [57], which indicates intermediately favorable conditions for decay. This index is similar to other reports regarding the SCI of different countries, including Hungary [63–66].

The decay rate can be indicated by similar reports that were carried out in this testing field. Untreated species usually start to grey after 2–3 months, which was hindered by heat-treatment (Turkey oak, hornbeam, Pannonia poplar [67], Scots pine [67,68], and beech [68]), heat-treatment in paraffin (beech and poplar [69]), oil-heat-treatment (poplar [70], beech and oak [71], beech [72], black locust [73]), impregnation with silica nanoparticles [74], and with iron nanoparticles (beech and Scots pine [75]).

In the study regarding heat-treating hornbeam, L* of heat-treated hornbeam increased, and a* and b* decreased after six months. The samples cracked, greyed, and the untreated ones even molded after two months. Wasp stripping became apparent on untreated samples after five months. Heat-treated hornbeam was more dimensionally stable and durable, but the color difference after weather exposure was greater than that of untreated hornbeam [67].

The highest ΔL* was experienced in the first months of weather exposure (spring and summer of 2018). In the first weeks, there was an initial increment, then L* decreased at different rates depending on the wood and coating type. After about 5–9 months, the biggest L* reduction was achieved in every sample. The greatest L* increment was in acetylated hornbeam samples, especially those that were coated with colorless coating (OO0, AS0, BS0) and BOT. Hornbeam samples experienced the biggest L* reduction, especially those coated with AOT and BS0. Dark pigmented coatings like ASU, BSP, and BSR had the lowest ΔL* (Figure 8). The brightening of acetylated hornbeam and darkening of natural hornbeam was the result of the photooxidation of lignin, hemicelluloses, and extractives, which corresponds to the literature [29,30].

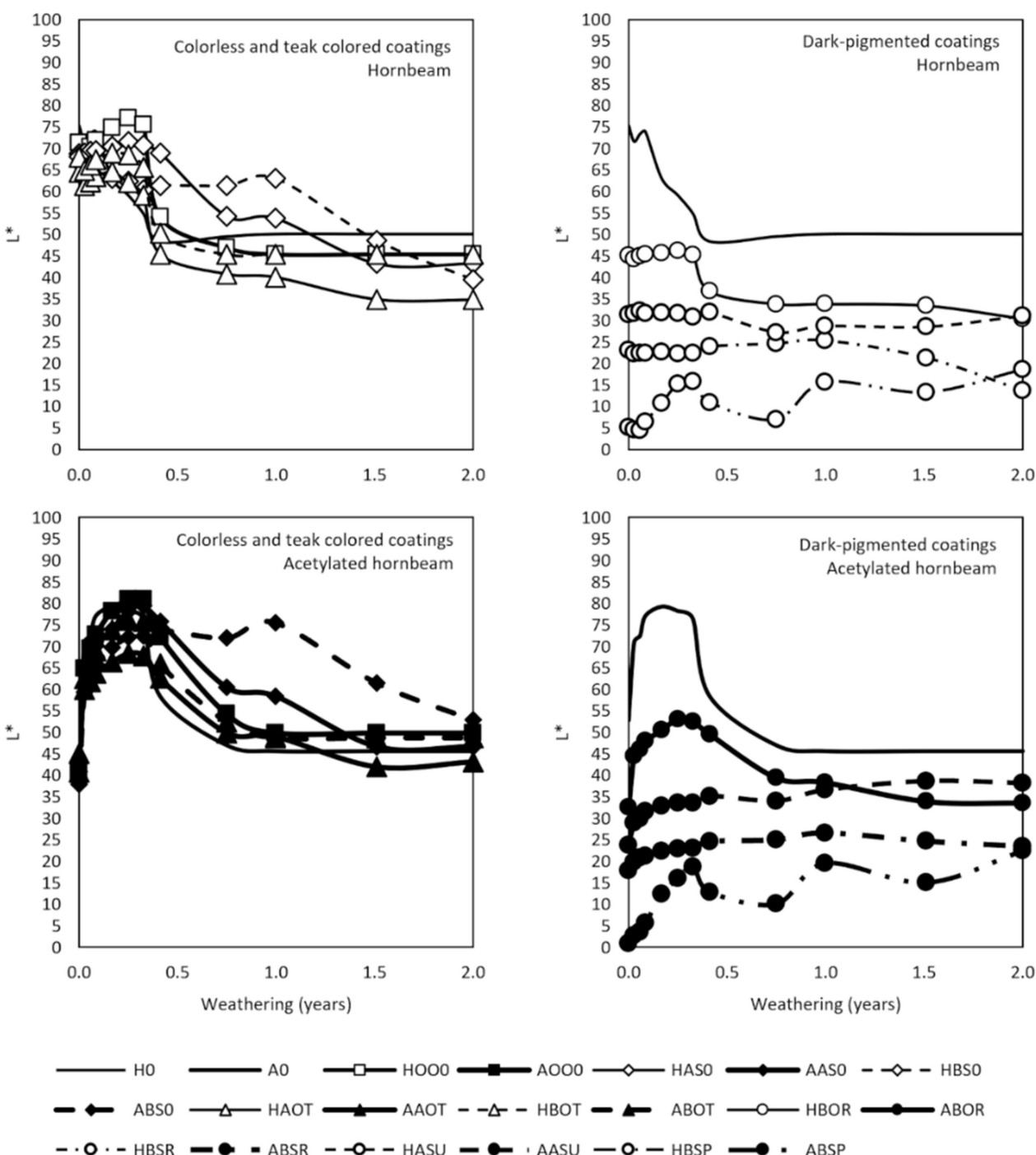

**Figure 8.** Change of lightness (L*) during two-year-long weather exposure.

In the first weeks, there was an initial increment of a* in hornbeam samples; before they started greying, their color shifted to a darker yellow. Then a* decreased at different rates depending on the wood and coating type. The greatest a* increment was in BS0 before the coating failed, and the surface began to grey as with the other colorless coatings. BSP coated acetylated hornbeam also had a great increment of a*. The biggest a* reduction was in BOR, BOT, and AOT. ASU had the lowest Δa* (Figure 9).

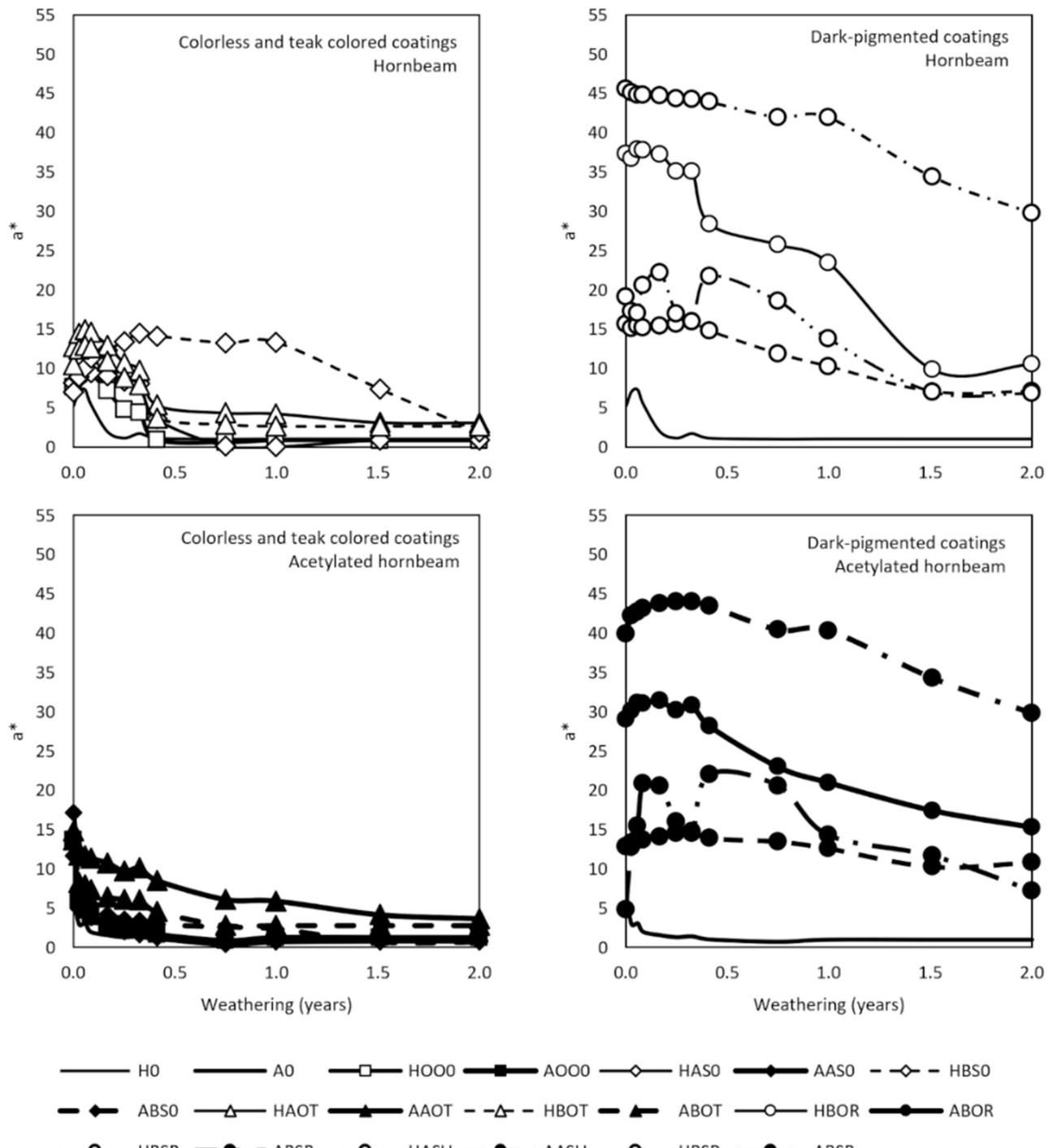

**Figure 9.** Change of red hue (a*) during two-year-long weather exposure.

Initial increment was also the case of b*, but only for natural hornbeam samples and not for acetylated samples. Among colorless coatings, BS0 maintained the wood color for the longest period, but after the coating failed, the surface began to grey, resulting in an even larger color difference than that of the uncoated wood. The b* of every coating–except for BSP and BSR–decreased over time during weather exposure. AOT, BOT, and BOR experienced the biggest reductions. ASU, BSR, and BSP had the lowest Δb* (Figure 10).

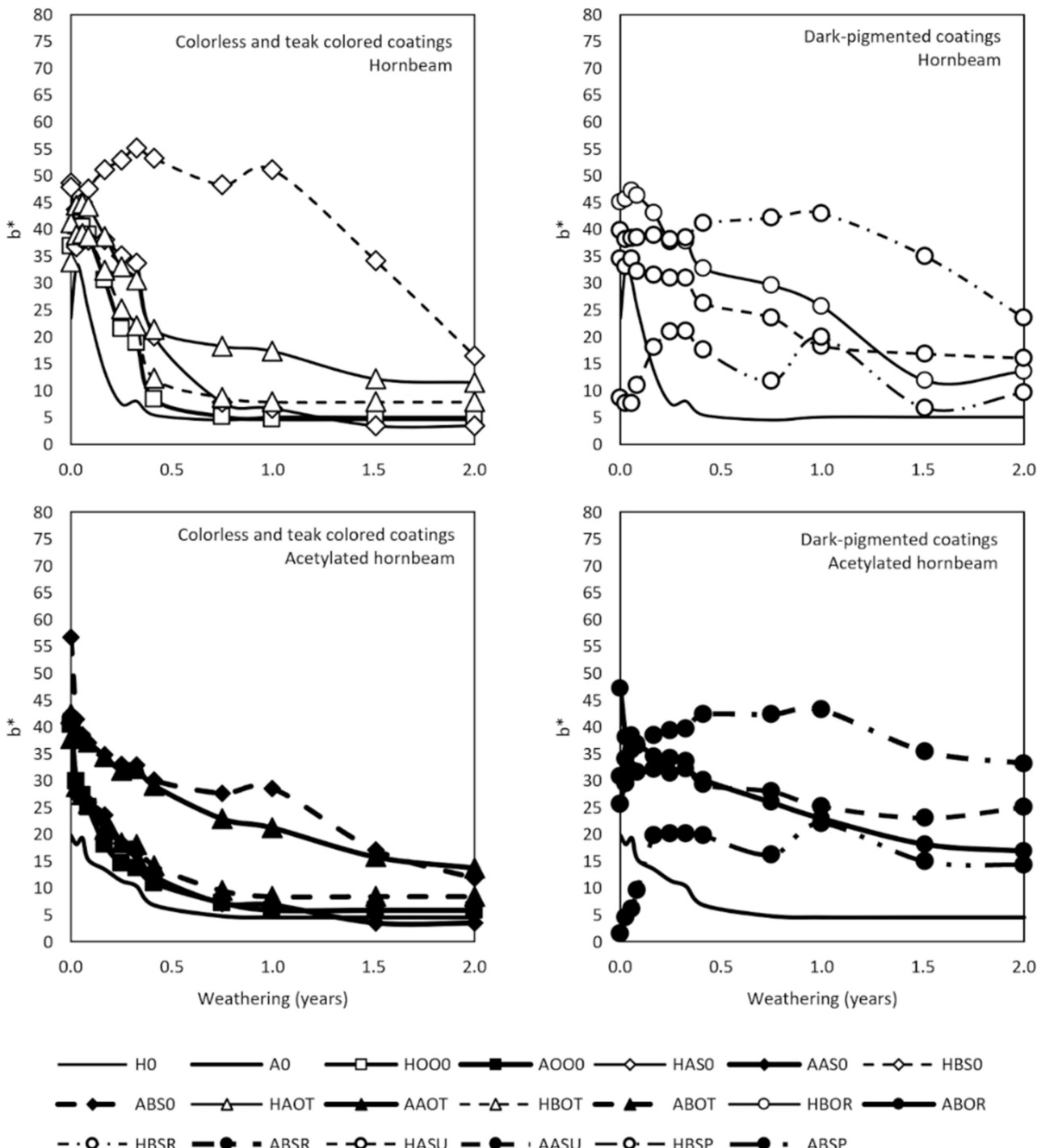

**Figure 10.** Change of yellow hue (b*) during two-year-long weather exposure.

Colorless coatings had a high rate of ΔE* in the first five months. At about 5–9 months, the surface started to grey, and the rate of change decreased greatly after one year. The greatest ΔE* in colored coatings was in BOR, BOT, and AOT. In most cases, coated acetylated hornbeam had lower ΔE* than hornbeam at the end of the exposure. Moreover, its trend curve was flatter. BSR, BSP, and ASU achieved the lowest ΔE* (Figure 11). S-Shaped and convex deterioration patterns can also be found in the diagrams [10].

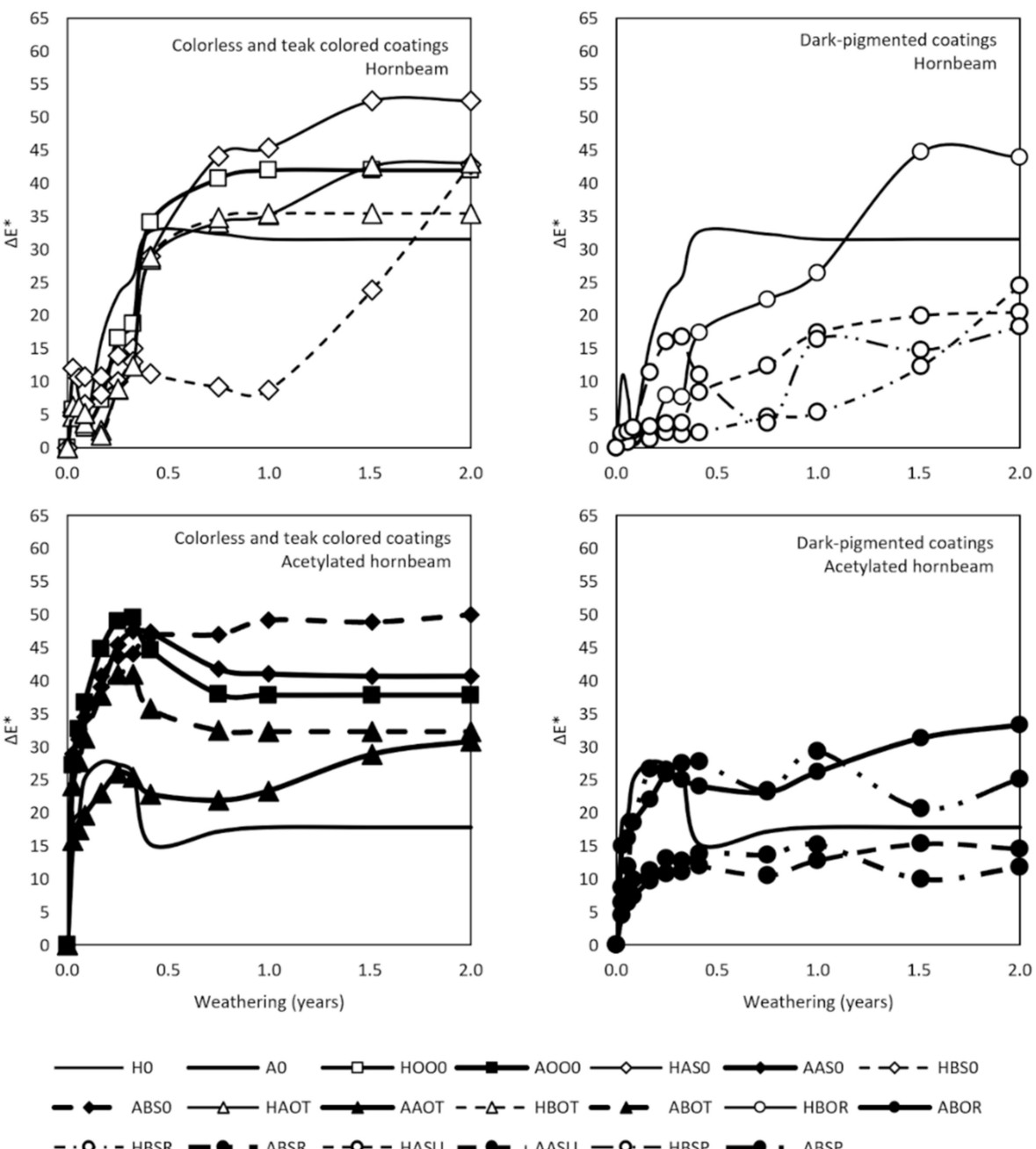

**Figure 11.** Change of color difference (ΔE*) during two-year-long weather exposure.

Color change in coating takes place due to photo oxidation, cross-linking, and leaching, where pigments and hydrophobic compounds are leached by precipitation. Light filters through clear coatings, which lead to photodegradation similarly to uncoated wood. During lignin degradation, free radicals are formed, which affect the wood-coating interface. Free radicals degrade the linkages between wood and coating, which can cause delamination while the coating itself is not degraded [20].

Table 6 demonstrates the initial ΔE* (compared to average color) or color variation in the wood before weather exposure and after one and two years of weather exposure. It shows that uncoated or coated wood already has a perceptible color difference or variation (greater than 2) before the weather exposure. This color difference can increase during photodegradation (wood degrades in spots, along cracks, wasp stripping occurs, coating fails, mold appears) but also decrease (even photodegradation resulting in complete greying).

**Table 6.** Color variation of weather-exposed hornbeam (color difference compared to average color). Average values of 20 measurements are presented.

| Marking | Hornbeam | | | Acetylated Hornbeam | | |
|---------|----------|--------|---------|---------------------|--------|---------|
|         | 0 Year   | 1 Year | 2 Years | 0 Year              | 1 Year | 2 Years |
| 0       | 2.22     | 3.68   | 3.68    | 6.07                | 18.08  | 18.08   |
| OO0     | 3.59     | 2.67   | 2.67    | 5.94                | 2.67   | 2.67    |
| AS0     | 2.86     | 2.73   | 3.91    | 5.30                | 3.12   | 2.73    |
| BS0     | 2.47     | 2.55   | 5.37    | 7.27                | 2.89   | 3.05    |
| AOT     | 3.05     | 3.54   | 6.20    | 2.80                | 4.08   | 5.78    |
| BOT     | 2.93     | 3.42   | 3.42    | 4.71                | 3.21   | 3.21    |
| BOR     | 3.24     | 3.92   | 10.31   | 3.00                | 5.00   | 6.30    |
| BSR     | 2.39     | 1.77   | 6.82    | 1.77                | 1.73   | 3.08    |
| ASU     | 2.08     | 4.12   | 2.13    | 2.70                | 3.31   | 2.15    |
| BSP     | 9.93     | 6.14   | 13.06   | 2.65                | 6.67   | 8.07    |

The results can be observed on the photo scans (Figure 12), and color chart (Figure 13). BSR and ASU were the most photostable coatings. Among the colorless coatings, BS0 was the most durable. BSP had better durability during only UV exposure, without other weathering factors. The brightening of acetylated hornbeam was very apparent. On the other hand, the coating of untreated hornbeam failed in many cases earlier than that of acetylated hornbeam, due to lower dimensional stability and durability.

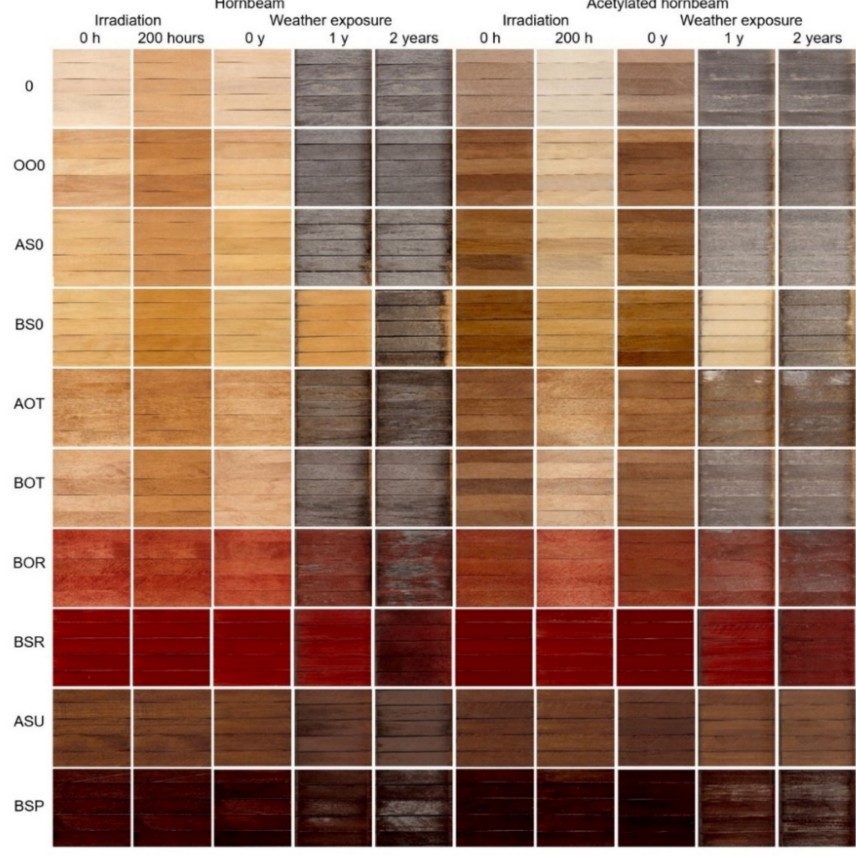

**Figure 12.** Scans of uncoated and coated hornbeam samples before and after 200-h-long irradiation and two-year-long weather exposure.

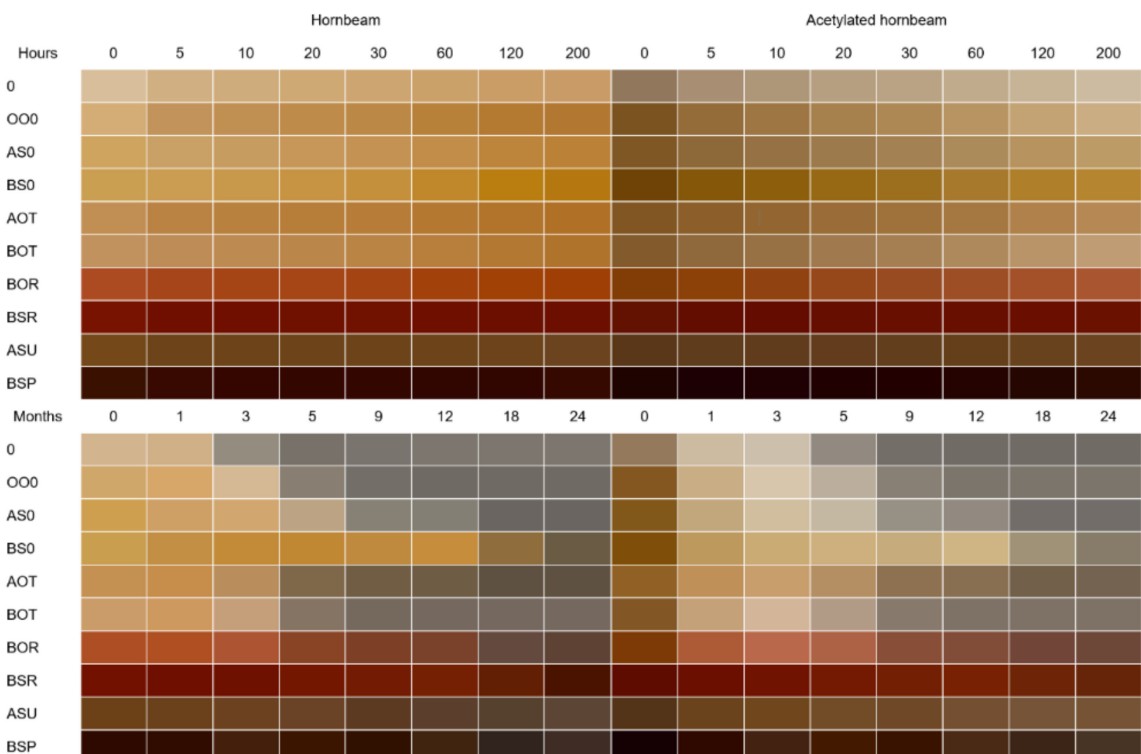

**Figure 13.** Color chart of uncoated and coated hornbeam and acetylated hornbeam samples before and during 200-h-long irradiation, and with two-year-long weather exposure.

The time course of the photodegradation process is listed below:

–   Uncoated hornbeam was the first to start greying among the tested samples–after about 21 days. Acetylated hornbeam started to grey after two months.
–   Coated acetylated samples started greying later than coated unmodified samples, which corresponds to the literature [37–39,41–46]. Black stain fungi can attack the delignified surface of acetylated wood [20,32].
–   OO0, AOT, and BOT started greying after three months due to no or low amounts of pigments, and not chemically bonding to the cell walls [24].
–   AS0 started to grey and crack after five months.
–   BS0 and BSP had a stable color for about one year and had a flaking surface after 1.5 years when the binder degraded, similarly to the literature [20,31].
–   BOR cracked after six months. It had a brightening interval until about five months, after which it starts to darken again while greying.
–   Cracks, flaking, chalking, and wasp marks became more apparent after nine months. Chalking was due to the photodegradation of the coating. The binder degraded where the film became thinner, and cracks appeared. Other researchers observed a similar process [20,21]. Wasps use the cellulose they strip from the wood surface to create nests. These marks were visible on almost all samples after one year of exposure.
–   After a shorter or longer period of time, the coating film thinned or peeled, and the natural process of color change took place as seen in literature [20,22].
–   BSR and ASU withstood the weather exposure for two years.

The results are comparable to those of mercury-vapor lamp irradiation [76]. Unlike the xenon lamp, the mercury vapor lamp emits light in all UV regions. Overall, 80% of its emission is UV light, from which 31% is the UV-A (380–315 nm) region, 24% is the UV-B (315–280 nm) region, and 25% is the UV-C (>280 nm) region. The xenon lamp emits UV light in only the UV-A region (400–340 nm). The average radiant power density of a mercury vapor lamp and a xenon lamp are 76 and 482 W/m$^2$, respectively. Comparing the xenon lamp results to the results of mercury vapor lamp irradiation reveals that

xenon lamp exposure had a greater impact on the color of both wood types. The reason for this is the radiant power density of the xenon lamp, which was more than six times higher. The color difference between one month of weather exposure and 200 h of mercury vapor lamp irradiation was 8.34 for uncoated hornbeam and 9.70 for uncoated and acetylated hornbeam.

FTIR studies confirmed that during the photodegradation of acetylated hornbeam caused by mercury-vapour lamp irradiation, the absorption of functional groups in lignin decreased. On the other hand, functional groups of methane, methylene, methyla, and carbonyl groups had greater absorption. The rate of degradation and structural changes were higher in acetylated hornbeam than untreated hornbeam, but the strengthening polymers did not degrade notably. Transformation and degradation of lignin and extractives resulted in the color brightening of acetylated hornbeam. This is confirmed by our results, where acetylated hornbeam had greater color difference values, but was more dimensionally stable, more durable, and less susceptible to structural changes caused by weathering [76].

Hornbeam a* and b* increased due to irradiation and decreased during weather exposure. This complicates comparisons. All the acetylated samples had the same or similar color after 200-h-long irradiation as one-month-long weather exposure, which suggests 3–4 times faster aging. The results of untreated hornbeam varied. In many cases, only L* was similar because a* and b* changed in different ways (Figure 14 and Table 7). A similar research study reported strong linear relationships between the first 70 days of natural sunlight to 60 h under artificial xenon light irradiation (180 W/m$^2$), suggesting a 30 times faster aging process in the artificial xenon light. After that, the linear relationship vanished [77].

Among the weathering factors, sunlight (UV radiation) causes the greatest color and surface changes. Irradiating with a mercury vapor lamp or a xenon lamp can help to study the photodegradation mechanism in wood, but they cannot simulate natural sunlight [17]. There are some high correlations with statistical significance between the in-service test (façade and decking cladding elements exposed to weather) and the laboratory test (exposed to blue stain fungi and xenon lamp irradiation), but it would be impossible to precisely predict the level of color change using only laboratory tests [18].

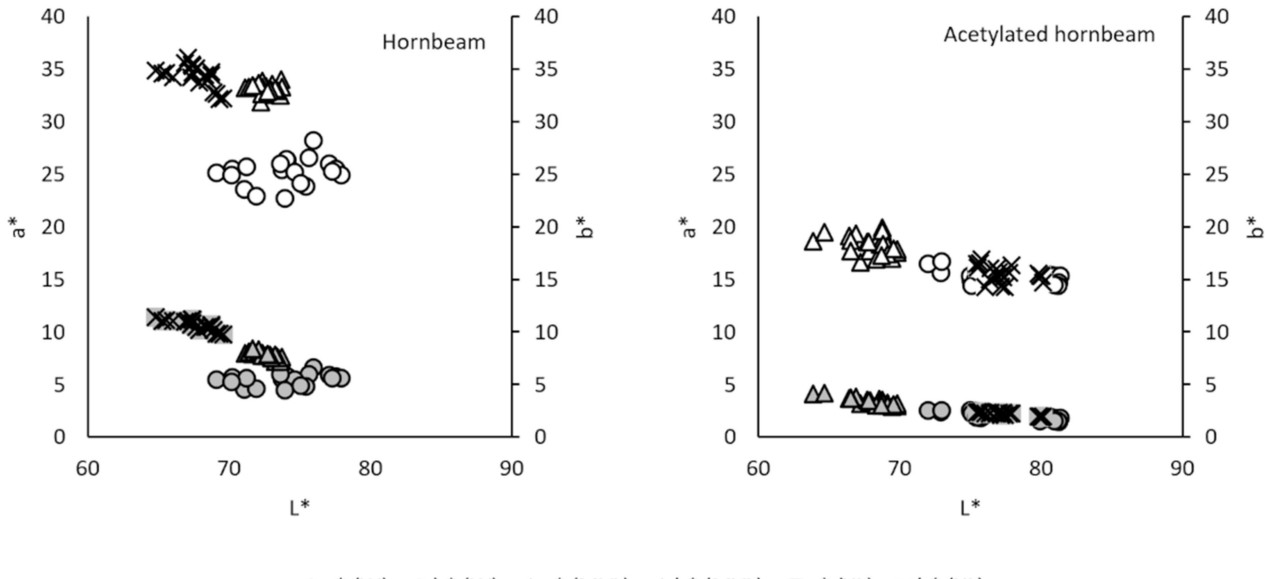

**Figure 14.** Comparison of color change (L*: lightness, a*: red hue, b*: yellow hue) of uncoated hornbeam and acetylated hornbeam caused by different photodegradation processes (W: weather, XL: xenon lamp, MVL: mercury-vapor lamp).

**Table 7.** Defining the length of weather exposure needed to achieve similar color change of 200-h-long xenon lamp irradiation.

| Marking | After 200 h of Xenon Lamp Irradiation | | | After Weather Exposure | | | | ΔE * |
| | L * | a * | b * | L * | a * | b * | Duration (Month) | |
|---|---|---|---|---|---|---|---|---|
| | | | | Untreated hornbeam | | | | |
| 0 | 67.59 | 10.64 | 34.27 | 73.98 | 5.46 | 25.20 | 0–1 | 12.24 |
| OO0 | 55.14 | 16.91 | 48.41 | 71.96 | 9.53 | 39.01 | 0–1 | 20.63 |
| AS0 | 58.92 | 15.52 | 49.36 | 69.52 | 9.47 | 38.05 | 0–1 | 16.64 |
| AOT | 53.41 | 18.82 | 50.86 | 63.45 | 14.64 | 44.32 | 0–1 | 12.70 |
| ASU | 31.93 | 15.14 | 31.48 | 31.66 | 15.18 | 32.12 | 1 | 0.69 |
| BS0 | 55.45 | 17.91 | 68.43 | 61.43 | 14.11 | 53.22 | 5 | 16.77 |
| BSP | 6.76 | 25.07 | 11.50 | 6.41 | 20.53 | 10.90 | 0–1 | 4.59 |
| BSR | 21.42 | 43.76 | 36.78 | 22.19 | 44.33 | 38.11 | 3 | 1.64 |
| BOT | 54.12 | 17.98 | 49.27 | 67.41 | 12.61 | 38.63 | 0–1 | 17.86 |
| BOR | 39.60 | 38.57 | 50.08 | 45.44 | 37.75 | 46.33 | 0–1 | 6.99 |
| | | | | Acetylated hornbeam | | | | |
| 0 | 77.32 | 2.20 | 15.51 | 77.15 | 2.01 | 15.35 | 1 | 0.30 |
| OO0 | 72.74 | 4.12 | 25.71 | 72.76 | 3.91 | 25.07 | 1 | 0.68 |
| AS0 | 66.39 | 5.29 | 32.76 | 70.19 | 3.83 | 25.97 | 0–1 | 7.92 |
| AOT | 60.41 | 11.12 | 36.13 | 63.76 | 11.36 | 37.21 | 0–1 | 3.53 |
| ASU | 32.21 | 14.09 | 31.19 | 31.69 | 13.72 | 31.54 | 1 | 0.73 |
| BS0 | 59.15 | 9.97 | 52.19 | 65.87 | 5.42 | 37.03 | 0–1 | 17.20 |
| BSP | 4.81 | 17.72 | 8.13 | 5.62 | 20.85 | 9.53 | 0–1 | 3.53 |
| BSR | 21.28 | 43.05 | 36.54 | 21.29 | 43.13 | 36.55 | 1 | 0.09 |
| BOT | 67.30 | 7.42 | 26.20 | 68.91 | 7.33 | 25.77 | 1 | 1.67 |
| BOR | 45.75 | 32.43 | 38.06 | 48.00 | 31.05 | 36.86 | 0–1 | 2.91 |

In this study, OO0, AS0, AOT, and ASU were linseed oil-based coatings; the rest were mainly polymer made from natural oils, fatty acids, and resins. According to artificial weathering studies, linseed oil has an early stage of photodegradation, where the oxidation of triglycerides, autoxidation of unsaturated fatty acid components, formation of conjugated unsaturations, and the development of extensive cross-linking takes place. In the next phase, the labile cross-links are consumed, and a network with higher stability is formed with unreacted triglycerides and low molecular weight molecules formed by fragmentation. After years of ageing, a progressive oxidation of the alkylic segments take place, leading to partial fragmentation and growth of oxygenated groups [23]. Linseed oil provides nutrition for mildew, which is especially discernable on hornbeam samples. In wet and damp areas, when coated with oil, acetylated hornbeam is likely to be attacked by mildew [28]. The high photostability of BS0, BSP, and BSR is probably due to its pigments, fillers, and additives like vegetable-based emulsifier, lead- and barium-free stabilizers and drying agents, methyl cellulose, film treaters (3-iodine-2-propynyl butylcarbamate (max. 0.2%), and 2-octyl-2H-isothiazol-3-one (<0.01%)). Their film thickness was also somewhat greater compared to other coatings, but this was not measured in this study.

## 4. Conclusions

According to the results of the present study, acetylation decreased the coating absorbance of hornbeam, which could lead to lower photostability properties. On the other hand, the coating system demonstrated better quality due to the lower dimensional stability of acetylated hornbeam. These findings corresponded to literature. Thus, longer service life is achieved, and less maintenance is required with the use of acetylated hornbeam wood.

The color of acetylated hornbeam is not photostable. As was the case during ultraviolet degradation, it also undergoes chemical changes that results in surface brightening and eventual greying outdoors. This was the result of the degradation and transformation of lignin and extractives, together with leaching caused by rain, and wasp stripping. Coating systems can hinder the effects of photodegradation, but long-lasting results require regular maintenance. Dark pigmented stains last longer and have a lower color difference compared to the initial color. However, they cover and hide the wood grain more than colorless and brighter pigmented coatings do.

Among the tested colorless coatings, Biopin colorless stain provided the most photostable coating film. Among all tested coatings, Auro umbra-colored stain and Biopin Swedish red-colored stain obtained the best results. We recommend using these coatings (or those with similar properties) to finish products made from acetylated hornbeam wood in order to have long-lasting color during its service life.

Color change was greater with the xenon lamp irradiation than with the mercury vapor lamp irradiation because the xenon lamp possessed higher radiant power density. In acetylated samples, 3–4 times faster aging was achieved, because the results after 200-h-long irradiation and after one month of weather exposure (April to May 2018) were the same or similar. Based on these findings, irradiating with a xenon lamp can help manufacturers and users predict the color change of acetylated hornbeam wood under similar exposure conditions as in this research.

**Author Contributions:** Conceptualization, F.F.; funding acquisition, F.F. and R.N.; investigation, F.F. and M.B.; methodology, F.F.; project administration, F.F. and R.N.; resources, F.F.; supervision, R.N.; validation, F.F., M.B. and R.N.; visualization, F.F.; writing—original draft, F.F.; writing—review & editing, M.B. and R.N. All authors have read and agreed to the published version of the manuscript.

**Funding:** Project no. TKP2021-NKTA-43 has been implemented with the support provided by the Ministry of Innovation and Technology of Hungary from the National Research, Development and Innovation Fund, financed under the TKP2021-NKTA funding scheme. This work was also supported by the ÚNKP-17-3-1 New National Excellence Program of the Ministry of Human Capacities in Hungary.

**Institutional Review Board Statement:** Not applicable.

**Informed Consent Statement:** Not applicable.

**Data Availability Statement:** Not applicable.

**Acknowledgments:** The authors would like to thank Sixay Furniture Ltd., Orange Ltd. and Stoki Ltd. for providing their products for research purposes. We also thank Ferry Bongers for his helpful, valuable insights and constructive critiques.

**Conflicts of Interest:** The authors declare no conflict of interest.

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
