# Peer review of "Photostability of Oil-Coated and Stain-Coated Acetylated Hornbeam Wood against Natural Weather and Artificial Aging"

_coatings, doi:10.3390/coatings12060817_

Round 1
Reviewer 1 Report
“Studying the Photostability of Oil-coated and Stain-coated Acetylated Hornbeam Wood, against Natural Weather and Artificial Aging” describes the results of a thorough study of the photostability of selected coatings on natural and acetylated hornbeam wood. The experiments were well planned and performed, and the results were described in detail. However, the manuscript is more a report than a scientific paper because it lacks a scientific discussion and explanations/hypotheses about the reasons for the differences observed between individual coatings performance. Therefore, I strongly suggest elaborating on the scientific side of the paper. Some more detailed comments can be found in the .pdf file attached.

Reviewer 2 Report
Review of the paper „Studying the photostability of oil-coated and stain-coated acetylated hornbeam wood against natural weather and artificial aging” submitted by Fodor et al. to the Coatings Journal.
In my opinion, the conducted research is justified. Research on the protection of wood and the use of finishing treatments to protect it or improve its aesthetics are still ongoing. Taking this into account, I conclude that the research topic is interesting. You can also notice the ecological aspect of the experiments. The paper is linguistically correct.
In my opinion the word “Studying” is unnecessary in the title. I think “The photostability of oil-coated (…) artificial aging” indicates enough information and sounds clear. But this is just a general suggestion to consider.
Abstract is informative. It consists of the most important information. The same applies to keywords which, in my opinion, should be listed in alphabetical order.
The introduction is too long, not very specific, it presents interesting information, however, they are usually general statements, and do not refer to specific research results that can be found in the literature. Some more specific comments below:
Line 34: “exposed to air” is rather unusual term. Did authors mean the outdoor or environmental conditions?
Paragraphs 3,4,5 (Line 39-57) consider the topics that I think could be omitted. They do not introduce the reader in any way to the topics discussed in the rest of the text. A description of the interpretation of results, etc., if intended to be included, is usually found in the methodology.
Line 64-74: If the authors intended to describe the climatic of Hungary, it might be worth referring to how this may affect the research on the weathering of samples and the properties of wood. At the moment, it causes some confusion for the reader and it is not entirely clear what the author's intentions were.
Line 86-96: In my opinion, this would be the part of the text that would be better suited to discussing the results.
Line 115: Authors emphasizes that such solutions are "convenient and less expensive". Maybe it is also worth mentioning at this point about coating’s leaching, their frequent mechanical damage and the need to renew the application.
Line 126 and 128: In first sentence there is an information that “(pigmented) stains may form a film layer” and in the next one: “Pigmented stains (…) are less effective than film forming paints” which is confusing and requires clarification.
Line 150: The information about the acetylated wood presented this way is just a general, already well-known statement, maybe it would be more interesting if this statements would be supported by the results of the experimental research.
Line 153-155: When presenting information on the development of fungi on the surface of acetylated wood, it may be more interesting for readers to focus on explaining why this occurs rather than presenting a general trend.
Line 165-177: I consider this paragraph the most valuable of the entire introduction.
Line 190: In the case of research on wood, it is worth to provide the Latin name of the species.
Line 200: The form of "10-10 samples" is unclear. Moreover, “… of each type of coating and species” What species authors had in mind?
Table 1: Maybe it would be clearer if the authors would include the name of the manufacturer in the table as a separate column than when they were included in the text.
Table 2: “2…3” is unclear. What is the reason for that specific amount applied in case of AOT?
Line 206-207: The information about the Department of Physics and Electronics at the University of Sopron is unnecessary in my opinion.
Line 214: The use of word “provided” is a little confusing. It sounds as if the experiment was based on some set conditions. It would be better to state that their job was to control and investigate the conditions during the experiment.
Line 241, 242: The abbreviations such as: RGB and sRGB require explanations.
Figure 2: What is the reason for giving the same data twice in different units?
Table 3: Were these values measured before further investigation of the effects of UV or outdoor conditions? As initial results where we cannot see changes, they can also be part of the methodology. Especially since they haven't even been commented on.
Figure 3-6 and 8-11 are unreadable. Figures are too small in general taking into account the number of variants etc. The thickness of the lines and the applied methods of marking the variant do not allow to distinguish the curves.
Line 284: The statement “significant changes” indicate the “statistical significance”. Maybe it is worth to write it differently, taking into account the lack of statistical analysis.
Table 5: The symbols for months, AVG, SUM etc. should be explained in the note below the table.
Line 339: Maybe it would be interesting to explain the reader why such effect occurred?
Line 364: What is the reason for this reference ([10])?
Line 372: The sentence “…but also decrease (even photodegradation, even, complete greying compared to colorful wood grain)” is unclear.
Line 390-399: Maybe it would be clearer to present this paragraph in the form of the table.
Line 427-429: Is this the observation based on the Table 7?
Conclusions: Line 431-439: Here you can find the general information about acetylated wood. I understand the author's intention, but perhaps it would be worthwhile to shorten this comment to the brief conclusion as it was not the aim of the research.
Line 443-444. Two treatments were listed as “the most photostable” and “the best results”. Maybe authors should pick one and call this the best.
Overall, I think it is an interesting paper and it can be a valuable reference. It regards the current topic and it has an ecological significance. This article is a good read, and my comments are often editorial. However, my main complaints concern the lack of statistical analysis (especially useful when dealing with variable conditions and non-homogeneous material) and the places where there is no scientific discussion of the results based on comparisons with the literature. In summary, I think it would be a great paper after minor corrections.
Reviewer 3 Report
coatings-1741508
This is a very well-written manuscript. The way the authors presenting the data is impressive. The paper can be accepted after the authors attend to these minor comments:
1. The introduction is well-written but it is a bit lengthy. I would recommend the author to shorten it.
2. Table 2, which do you mean by 2…3?
3. Line 208 – 482W/m2, please put a space between number and unit. Please check the others throughout the manuscript.
4. Table 3 – is the values presented in the table an average of 20 readings? Please mention clearly. Same goes to Table 6.
5. Line 259 – “had the smallest of all coatings…” smallest of what?
6. Line 290-291 – The way the authors describe seems like a* and b* are positive correlated with each other, is that so? a* increased as b* increased? Please clarify.
7. Discussion is lacking throughout the manuscript, particularly on the difference between each type of coating and what make them change colour at different extent.
Round 2
Reviewer 1 Report
The manuscript has been thoroughly corrected. I recommend it for publication.